# The Zero-Shot Illusion in the Wild:
# Diagnosing Boundary Failures in AIGC Detection

### Abstract

Detecting AI-generated images remains a significant challenge due to the rapid proliferation of diverse generative models and workflows. While many detectors claim strong zero-shot generalization in curated benchmarks, they struggle to generalize to the more complex distribution of real-world AI images, where outputs are from workflows combining multiple models, LoRAs, adapters, post-processing, etc. We expose this *zero-shot illusion* with **WildGen**, an in-the-wild benchmark built from 342K community-created AI images spanning 342 sources, together with 347K real images from 8 authentic datasets. On WildGen, detectors that perform well on existing benchmarks exhibit source-dependent failures: 11 of 12 training-based methods remain below 58% balanced accuracy, and the best training-free method reaches only 75.21% AUC. The generalization gap arises primarily from decision boundary misalignment rather than a fundamental deficiency in feature representation. We demonstrate that adapting off-the-shelf detectors to unseen models yields 97.22% balanced accuracy (95.77% if only 18 examples per generator are used) for AI-vs-real classification, and real-source identity is predictable at 84.33% balanced accuracy. Detectors trained on WildGen can also be adapted to existing benchmarks such as GenImage and ForenSynths using few-shot target data. These results suggest that, rather than seeking a static universal AIGC detector, a more effective and realistic path is continuous few-shot adaptation that maintains detector boundaries as generators and content-creation workflows evolve.

## 1 Introduction

Rapid advancements in generative models like FLUX.2 (Black Forest Labs, 2025), Qwen-Image (Wu et al., 2025a), and Gemini (Team et al., 2023) now yield high-quality images that are often indistinguishable from reality. While this progress supports creativity, it also increases the risks of disinformation, impersonation, and provenance laundering. Robust AI-image detection therefore requires more than a high score on a static benchmark: it requires evidence that a detector's claimed generalization survives the long-tail, user-driven conditions where generated images actually circulate.

Existing defense mechanisms, categorized into training-based and training-free approaches, have reported high success rates on standard benchmarks like ForenSynths (Wang et al., 2020) and GenImage (Zhu et al., 2023). We argue that these results create a *zero-shot illusion*: methods appear to generalize because the benchmark distribution is narrow or shares shortcuts with the training data, but their decision rules do not survive a truly heterogeneous wild setting. In our evaluation, several detectors collapse toward one class or depend on source-specific artifacts when exposed to community generators and traceable real-image sources. This failure reveals that the central question is not whether a detector is universally zero-shot, but where its learned *decision boundary* is valid, how source bias affects it, and when that boundary must be updated.

The generative ecosystem itself is far from static. Real-world generation pipelines have diversified to include granular modifications of base checkpoints, model merging, and modulation via Low-Rank Adaptations (LoRAs) (Hu et al., 2022). Images are rarely the raw output of a single model; they are the result of workflows involving text prompts, prompt embeddings, adapters, and post-processing (Figure 1). At the same time, the "Real" side of a detector is not homogeneous: real-image datasets have their own source-specific semantics,

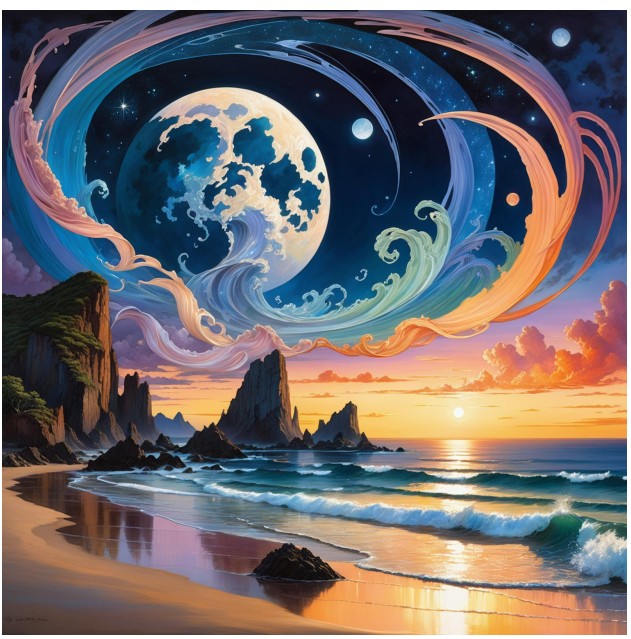

| Pipeline | [Base] SDXL Unstable Diffusers - YamerMIX · [LoRA] Detail Tweaker XL:0.85, extremely detailed:1.0 · [Embeds] Fast Negative Embedding, Beyond SDXL Negative |
|---|---|
| Prompt | In this mythical night scene viewed from high above, the panorama unfolds with breathtaking grandeur... *[truncated]* |
| Neg. Prompt | FastNegativeV2, low res, poor quality, BeyondSDXLv3 |
| Config | Steps: 40 · CFG: 6.0 · Sampler: DPM++ 3M SDE Karras · Seed: 4004999179 |

Figure 1: A representative example illustrating the composite nature of the benchmark. The generation pipeline involves a complex combination of a base checkpoint and multiple LoRAs, modulated by prompt embeddings. This workflow demonstrates the high complexity of prompts and resource chaining in real-world content creation.

resolutions, compression patterns, and curation biases. A benchmark that ignores either side of this boundary can overstate progress.

To study these effects, we introduce WildGen, a large-scale in-the-wild benchmark constructed from community-generated images and traceable real-image sources. The current corpus indexes 1,005,490 AI images from 447 mapped generative sources and 347,207 real images from eight identifiable real-data sources. The benchmark split includes 342 AI sources with at least 1,000 images each, reserving 100 validation images per AI source and 1,000 validation images per real source. This design lets us expose zero-shot failure in the wild, diagnose whether it comes from feature deficiency or boundary misalignment, and test how efficiently the boundary can be maintained under cross-benchmark, real-source, and environmental shifts.

Our contributions are summarized as follows:

- We introduce WildGen, an in-the-wild benchmark for AIGC detection that better reflects both sides of the deployment distribution. On the AI side, WildGen contains community-generated images produced by real workflows involving base checkpoints, LoRAs, prompt engineering, and post-processing. On the Real side, it uses 347k traceable images from eight diverse sources, reducing the risk that detectors obtain apparent generalization by overfitting to a narrow real-image distribution.

- We benchmark representative existing detectors on WildGen and expose a zero-shot illusion in current AIGC detection. Methods that report strong performance on curated benchmarks often fail on heterogeneous community generators and diverse real sources: most training-based detectors remain near chance or collapse toward one class, and the best training-free method reaches only 75.21%

AUC. Source-dependent false positives reveal an additional real-side failure mode: detectors trained or calibrated on narrow real datasets can misclassify unfamiliar real sources as fake.

- We show that few-shot adaptation is a practical mechanism for maintaining detector boundaries under distribution shift. Within WildGen, frozen CLIP features already support strong AI-vs-real separation with very few examples per generator, while cross-benchmark transfer to GenImage and ForenSynths reveals larger boundary mismatch that benefits from target-domain adaptation. The best adaptation strategy depends on the distance of the target distribution and the capacity of the detector, rather than following a single universal recipe.

## 2 Related Work

### 2.1 AI-Generated Image Detection

Existing methods can be categorized into two primary paradigms: *training-based* and *training-free* approaches (Table 1).

**Training-based Methods.** Early detection methods primarily targeted CNN-based generators. Wang et al. (2020) demonstrated that binary classifiers (e.g., ResNet-50, He et al., 2016) trained on specific GANs could generalize to unseen models through robust data augmentation. This sparked research into cross-generator generalization, including gradient-domain analysis (LGrad, Tan et al., 2023) and multi-branch architectures (HiFi-Net, Guo et al., 2023). With the rise of diffusion models, newer strategies have emerged: DIRE (Wang et al., 2023) exploits reconstruction error as a discriminative clue, RINE (Koutlis & Papadopoulos, 2024) and FatFormer (Liu et al., 2024b) leverage foundation model features (e.g., CLIP, Radford et al., 2021) with specialized adapters, and NPR (Tan et al., 2024) targets upsampling artifacts via nearest-neighbor representations.

**Training-free Methods.** Recent works explore zero-shot detection by leveraging intrinsic properties of pretrained foundation models without fine-tuning. One direction exploits feature robustness: WaRPAD (Choi et al., 2025) and RIGID (He et al., 2024b) observe that real images maintain consistent DINOv2 (Oquab et al., 2023) representations under perturbations (e.g., wavelet removal or Gaussian noise), whereas AI-generated images exhibit instability. Another perspective leverages reconstruction fidelity: AEROBLADE (Ricker et al., 2024) measures reconstruction error through latent diffusion VAEs, finding that generated images, originating from the same latent space, reconstruct with significantly lower error than real photographs. Furthermore, ManifoldBias (Brokman et al., 2025) analyzes manifold biases by examining Stable Diffusion's UNet response to latent noise. While versatile, these methods often incur high computational overhead and, as we show in Section 4, struggle with the evolving artifacts of modern generators.

Table 1: Summary of open-sourced AI image detection methods. Training-based methods require labeled real/fake pairs, while training-free methods use pretrained features directly.

| Method | Type | Backbone |
|---|---|---|
| CNNDetection | Training | ResNet-50 |
| UnivFD | Training | CLIP ViT-L/14 |
| LGrad | Training | ResNet-50 |
| HiFi-Net | Training | HRNet |
| DIRE | Training | ResNet-50 |
| RINE | Training | CLIP ViT-L/14 |
| NPR | Training | ResNet-50 |
| AEROBLADE | Free | SD VAE |
| FatFormer | Training | CLIP ViT-L/14 |
| AIDE | Training | ResNet-50+ConvNeXt |
| RIGID | Free | DINOv2 ViT-L/14 |
| WaRPAD | Free | DINOv2 ViT-L/14 |
| DeeCLIP | Training | CLIP ViT-L/14 |
| Effort | Training | CLIP ViT-L/14 |
| ManifoldBias | Free | SD+CLIP+LLaVA |
| AlignedForensics | Training | ResNet-50 |

Table 2: Benchmarks for AI-generated image detection. AI Generators lists the generative models used to create synthetic images.

| Benchmark | Size | Real Source | AI Generators |
|---|---|---|---|
| ForenSynths | 720K | LSUN | ProGAN, StyleGAN, StyleGAN2, BigGAN, CycleGAN, StarGAN, GauGAN, Deepfake |
| DiffusionForensics | 100K | ImageNet, LSUN, CelebA-HQ | ADM, DDPM, IDDPM, LDM, PNDM, Stable Diffusion, VQ-Diffusion |
| GenImage | 1.3M+ | ImageNet | Midjourney, Stable Diffusion, ADM, GLIDE, Wukong, VQDM, BigGAN |
| Chameleon (Yan et al., 2024a) | 10K | Online sources | Professionally edited AI images |
| WildGen (Ours) | 342K AI + 347K real | 8 traceable sources | 342 diverse sources |

## 2.2 Benchmarks

Benchmarks have transitioned from early GAN-based collections like ForenSynths (Wang et al., 2020) to modern diffusion-based sets, including DiffusionForensics (Wang et al., 2023) and GenImage (Zhu et al., 2023). As summarized in Table 2, existing benchmarks typically exhibit two fundamental flaws. First, they often suffer from a "closed-set" bias by focusing on a limited number of standard architectures instead of the "long-tail" distribution of community-tuned models. Second, they frequently rely on legacy real datasets like ImageNet or LSUN, creating a "real-world gap" due to historical compression and sampling artifacts. We introduce WildGen to address these limitations through a multi-modal, community-driven distribution of high-fidelity synthetic and real data.

## 2.3 Emerging Paradigms and Perspectives

**From Zero-shot to Few-shot.** The proliferation of few-shot generation algorithms (He et al., 2024a; Zhao et al., 2023) has accelerated the customization of generative models, making few-shot adaptation increasingly vital for tracking evolving threats. Recent strategies like FTNet (Yao et al., 2025), FAMSeC (Xu et al., 2024), OmniDFA (Wu et al., 2025b), and OCC-CLIP (Liu et al., 2024a) rely on retrieval caches, learnable prompts, or parameter-heavy feature-reshaping, typically validated on small-scale benchmarks. Similarly, Urueña et al. (2025) utilize supervised contrastive learning followed by k-nearest neighbor classification to address open-set detection and attribution. In contrast, we target a heterogeneous long-tail of community generators and evaluate when lightweight boundary updates are sufficient, when full fine-tuning is more effective, and when zero-shot transfer remains reliable.

**Large Multimodal Models for Detection.** Some works (Li et al., 2025; Ji et al., 2025) have attempted to use Large Multimodal Models (LMMs) for detection. However, their utility is constrained by computational cost and the difficulty of reproducible benchmarking as hosted models are continuously updated. Appendix B provides a qualitative case study illustrating how general-purpose assistants reason about image provenance, but we do not treat it as quantitative evidence.

**Data-Centric Detection.** Finally, our work connects to the data-centric AI movement (Gadre et al., 2023). We posit that the "Generalization Gap" in detection is often a boundary and data-coverage problem rather than purely an architectural failure. By shifting the focus from static models to measured distribution shift and targeted updates, we emphasize that maintaining robustness requires traceable data, real-source controls, and adaptation protocols rather than claims of static universality.

## 3 WildGen Benchmark

To construct a benchmark that faithfully reflects the complexity of the current generative landscape, we implemented a multi-stage data acquisition and curation pipeline targeting a leading community-driven model-sharing platform **Civitai** (Civitai, 2025). To promote reproducibility and future research, the WildGen benchmark is included in the Supplementary Material and will be made publicly available together with all associated code upon publication.

### 3.1 Data Acquisition and Model Definition

We systematically scanned the platform to harvest a diverse collection of AI-generated images. For each sample, we extracted available generation metadata, including positive and negative prompts, and the detailed model configuration used. Unlike simple templates, these prompts reflect intricate user engineering (e.g., negative embeddings, weighting), capturing the complexity of real-world usage (see Appendix A for additional pipeline examples).

To reflect authentic detection challenges, we define a "generative source" as the underlying base checkpoint. In modern modular generation pipelines, images are frequently conditioned by auxiliary components such as LoRA modules, ControlNet (Zhang et al., 2023) guidance, or *generative post-processing* utilities (e.g., ADetailer, Bing-su, 2023). We systematically treat these modifications as intra-class variations, assigning all images generated from the same foundational checkpoint to a unified source label. It is important to distinguish this *generative* post-processing–which occurs within the model's workflow–from the *distributional artifacts* (e.g., JPEG compression, resizing) that images undergo after publication; while WildGen captures both, our analysis focuses on the former's impact on architectural fingerprints. This aggregation strategy serves two critical purposes: it prevents label fragmentation in the benchmark and compels detectors to capture stable, architectural fingerprints rather than overfitting to transient stylistic or structural artifacts. Consequently, WildGen provides a more rigorous test of detection robustness against the heterogeneous and evolving landscape of generative modeling.

### 3.2 Two-Stage Curation Pipeline

To ensure that WildGen remains representative of the most relevant generative threats, we implement a two-stage curation pipeline. First, we filter for image-level quality by exclusively retaining samples that meet a strict verifiable community engagement threshold (likes + comments $\geq 1$). A prompt vocabulary analysis (Figure 2) reveals a dominance of quality-boosting tags such as "masterpiece" and "best quality", reflecting the aesthetic preferences of the community. This engagement-based filtering ensures that the dataset aligns with prevalent community trends, prioritizing the high-quality regime where AI content is most abundant and thus where the potential for confusion with authentic imagery is greatest. Regardless of this stylistic clustering, the semantic content remains diverse, covering a broad spectrum of subjects.

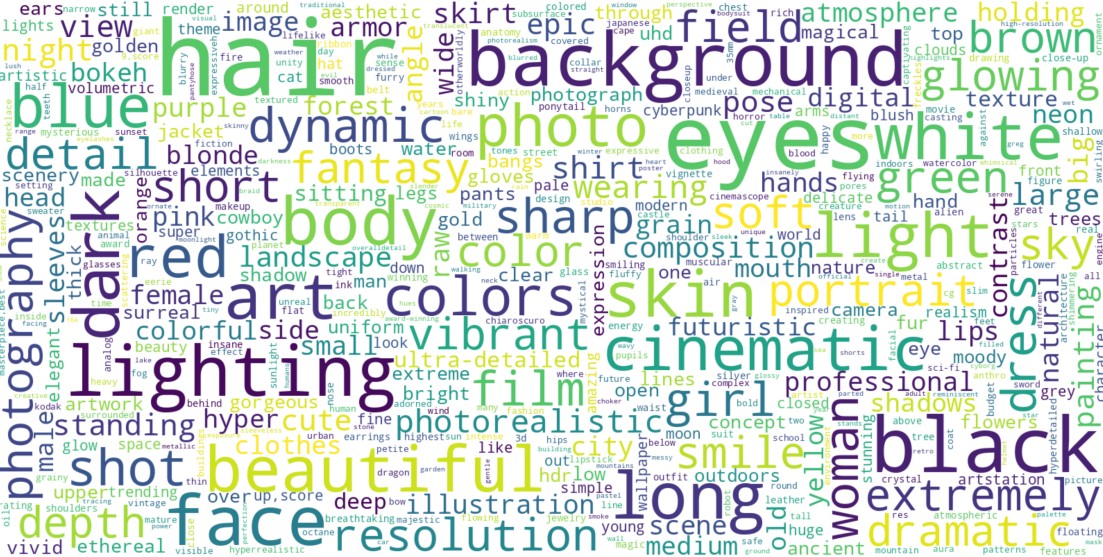

Figure 2: Word frequency analysis of the prompts in WildGen. The prominence of quality-related modifiers reflects the benchmark's focus on high-quality generations, which represent the most prevalent and challenging detection scenarios.

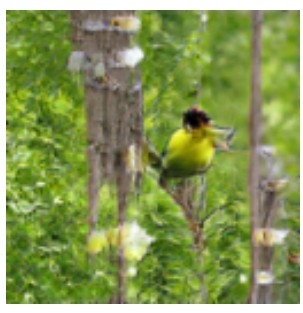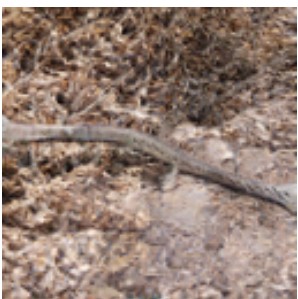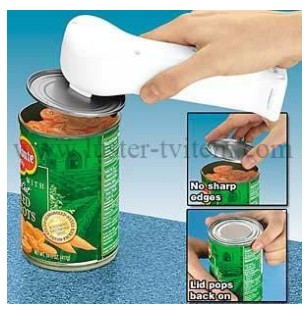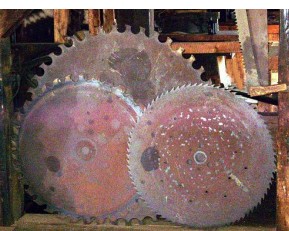

(a) GenImage (AI-generated)                    (b) ImageNet (Real)

Figure 3: Samples from GenImage (left) and ImageNet (right) often exhibit low resolution ($< 256$px) and significant compression artifacts, which detectors can exploit as "shortcuts" rather than learning robust discriminative features.

Second, we filter for **model-level significance** by identifying generators that have produced at least 1,000 such samples. For each eligible source, we deterministically select 1,000 images for the benchmark split to avoid distribution imbalance while preserving a larger indexed corpus for metadata analysis. The resulting split contains 342 eligible AI sources spanning different creators, base families, and aesthetic goals (the full catalog is provided in Appendix D), preserving the long-tail structural diversity required to test detector generalization.

### 3.3 Benchmark Statistics and Split

We initially collect 1,005,490 raw AI-image records from 447 mapped generative sources. We then apply the model-level eligibility criterion above, retaining only sources with at least 1,000 indexed images, and sample exactly 1,000 images from each source. After this filtering and balanced sampling procedure, 342 sources remain in the benchmark used in the following experiments, yielding 342,000 AI images: 34,200 for validation and 307,800 for the training pool. The split is shared across binary detection and source-identification evaluations. To keep WildGen aligned with the current generative landscape, it also includes recently released closed-source generators such as OpenAI 4o Image Gen, Imagen 4, and Nano Banana [1].

### 3.4 Real Data Integration

Another critical limitation in existing benchmarks is sampling bias within the "Real" class. Liu & He (2024) demonstrates that dataset bias persists even in billion-scale regimes; models can identify image sources (e.g., LAION vs. YFCC) with $>85\%$ accuracy by exploiting systematic semantic and stylistic differences. Thus, the distribution of the "Real" class is as critical as that of the "AI" one. When existing benchmarks rely exclusively on legacy datasets like ImageNet (Russakovsky et al., 2015) or LSUN (Yu et al., 2015), detection models risk learning dataset-specific artifacts (see Figure 3) rather than universal properties of authentic imagery. Consequently, these detectors often treat high-quality real images in the wild as out-of-distribution data. This is particularly problematic for training-free methods that calibrate detection hypotheses on specific real distributions; as confirmed in Section 4, these "universal" physical rules often fail when applied to our higher-fidelity real-world distribution.

We therefore replace the earlier unclear-provenance real subset with a traceable real-image index containing 347,207 images from eight sources: OpenImages (Kuznetsova et al., 2020), YFCC/Flickr CC (Thomee et al., 2016), Places365 (Zhou et al., 2017a), Unsplash (Unsplash), SUN397 (Xiao et al., 2010), ADE20K (Zhou et al., 2017b), Food101 (Bossard et al., 2014), and Google Landmarks v2 (Weyand et al., 2020). For each source, we reserve 1,000 validation images and use the remainder as a training pool, yielding 8,000 real validation images and 339,207 real training images. In binary evaluations, real images are pooled as the authentic class; in source-aware controls, each real source is represented as a separate class. This design lets us measure how much of a detector's boundary is driven by authenticity versus real-source identity.

---

[1] Unlike the rest of the AI corpus, the Nano Banana subset is drawn from the public HuggingFace dataset `bitmind/nano-banana`

# 4 Exposing the Zero-Shot Illusion

To ensure valid comparisons and reproducibility, we strictly evaluate representative methods with official open-source implementations and pretrained weights. Concurrent works lacking public inference code or checkpoints are excluded from this benchmark study. Evaluation uses the full fixed validation split: **100 AI images for each of the 342 eligible sources** and **1,000 real images for each of the eight real sources**. Despite reported state-of-the-art performance on standard closed-set benchmarks, existing methods across multiple paradigms show substantial boundary mismatch on the WildGen long-tail. This gap is the empirical basis of the zero-shot illusion: benchmark-reported generalization does not necessarily transfer to wild generators and controlled real-image sources.

## 4.1 Training-based methods

We categorize the evaluated training-based methods into three structural paradigms: (1) **Frozen Backbone Adapters** (UnivFD, Ojha et al., 2023; RINE, Koutlis & Papadopoulos, 2024; DeeCLIP, Keita et al., 2025; FatFormer, Liu et al., 2024b; Effort, Yan et al., 2024b), which adapt a fixed foundation-model backbone with lightweight trainable components (linear heads, adapters, or low-rank/orthogonal updates); (2) **Fully Trainable Classifiers** (CNNDetection, Wang et al., 2020; LGrad, Tan et al., 2023; DIRE, Wang et al., 2023; NPR, Tan et al., 2024; AlignedForensics, Rajan et al., 2025), which fine-tune backbones end-to-end; and (3) **Hybrid/Multi-Branch Architectures** (HiFi-Net, Guo et al., 2023; AIDE, Yan et al., 2024a) that fuse multiple streams. This categorization allows us to assess whether generalization capability correlates with specific architectural choices; for instance, whether large-scale pretraining (CLIP) offers an advantage over specialized training from scratch. Our evaluation shows that methods across all paradigms struggle to generalize to unseen models, as shown in Table 3.

Observing Table 3 and the score distributions in Figure 4b, we identify a distinct pattern in failure modes indicative of boundary overfitting. Methods tend to collapse toward a single class prediction when their learned boundary is misaligned with the target distribution. Several detectors achieve high recall on Real images but fail on unseen AI models, effectively defaulting to the "Real" label when encountering unfamiliar artifacts. Conversely, DIRE and Effort (Yan et al., 2024b) exhibit the opposite pathology: they correctly identify most AI images (DIRE: 99.3%, Effort: 94.2%), but their Real recall drops sharply (DIRE: 5.6%, Effort: 60.9%). These failures show that the zero-shot illusion is not a small calibration error: many static detectors memorize source-specific artifacts from their training sets rather than learning a boundary that transfers across real and generative distributions. The final two columns (Worst Source / Worst FPR) further show that these real-side errors are often concentrated on a few specific real sources—a shortcut we analyze in Section 5.4.

However, feature-space analysis shows that this failure is not simply caused by an absence of discriminative information. Frozen CLIP features support strong separation within WildGen, as visualized by the t-SNE projection in Figure 4a: a binary logistic regression probe reaches 98.15% balanced accuracy, and a 343-way kNN classifier (342 AI sources plus a single pooled Real class) reaches 89.06%/95.56% top-1/top-5 accuracy. Consequently, many observed failures are better understood as misaligned decision boundaries that do not cover the target distribution. This finding motivates the adaptation and generalization studies in Section 5, where we test when lightweight realignment is sufficient and when larger target-domain shifts require more capacity.

## 4.2 Training-free methods

Training-free methods operate by leveraging the intrinsic properties of foundation models, typically building upon hypotheses related to feature robustness or reconstruction fidelity (see Section 2). While these approaches eliminate the need for task-specific training and offer theoretical generalization, their practical efficacy on modern, high-fidelity community models remains largely unvalidated. Because these methods produce scores without a fixed decision threshold, we evaluate them with threshold-independent metrics (per-source balanced AP/AUC; Table 4) rather than the pooled balanced accuracy used for training-based detectors, so the two families' numbers are not directly comparable.

Table 3: Performance of training-based detectors on the full fixed WildGen zero-shot validation split: 34,200 AI validation images (100 per eligible source) and 8,000 real validation images (1,000 per real source). We report AI recall, Real recall, and the real source with the highest false-positive rate (FPR). The worst-source column highlights whether real-side errors are source-concentrated.

| Method | Bal. Acc. (%) ↑ | Rec-AI (%) ↑ | Rec-Real (%) ↑ | Worst Source | Worst FPR (%) ↓ |
|---|---|---|---|---|---|
| DIRE | 52.45 | 99.33 | 5.58 | SUN397 | 99.90 |
| Effort | 77.54 | 94.16 | 60.91 | Unsplash | 97.60 |
| NPR | 43.41 | 17.46 | 69.36 | YFCC/Flickr CC | 76.00 |
| HiFi-Net | 46.86 | 2.88 | 90.84 | YFCC/Flickr CC | 17.40 |
| FatFormer | 50.05 | 3.48 | 96.62 | YFCC/Flickr CC | 9.30 |
| AlignedForensics | 56.87 | 14.28 | 99.46 | YFCC/Flickr CC | 0.90 |
| DeeCLIP | 48.78 | 2.53 | 95.03 | YFCC/Flickr CC | 16.80 |
| LGrad | 44.62 | 6.19 | 83.04 | YFCC/Flickr CC | 45.30 |
| UnivFD | 49.99 | 0.87 | 99.11 | Places365 | 1.80 |
| RINE | 49.81 | 0.76 | 98.86 | Unsplash | 3.60 |
| CNNDetection | 49.69 | 0.16 | 99.22 | Unsplash | 2.20 |
| AIDE | 57.80 | 48.65 | 66.94 | YFCC/Flickr CC | 76.60 |

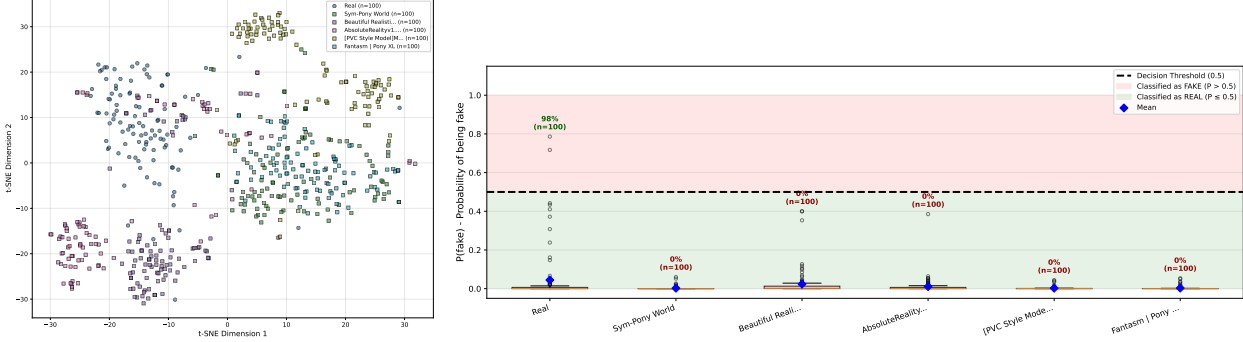

(a) t-SNE projection using CLIP features.  (b) Score distribution of UnivFD's Linear Probe

Figure 4: Visual analysis of UnivFD's performance on WildGen

Taking RIGID (He et al., 2024b), the highest-performing method (75.21% AUC; Table 4) in our evaluation, as a case study, its core premise relies on the fragility of AI-generated content to minor Gaussian noise perturbations in the DINOv2 feature space. This hypothesis was originally established using LDM (SD1.5) (Rombach et al., 2022) as the representative model to demonstrate a sharp drop in feature similarity under noise. However, as illustrated in Figure 6, the distribution of scores for AI and real images heavily overlaps on our benchmark, indicating that this "fragility" is not universal. Notably, Table 5 reveals a stark architectural disparity: the top-3 best-detected models are all fine-tuned from SD1.5, while the bottom-3 are based on SDXL (Podell et al., 2023).

To investigate this disparity, we conducted a spectral frequency analysis (Figure 5) to identify the specific artifactual fingerprints of the best-detected (qchan Mix v2.0, SD1.5-based) and worst-detected (Pixel Art Diffusion XL, SDXL-based) models. To isolate these artifacts, we computed a canonical "Real" spectrum by averaging the log-magnitude FFTs of 100 randomly sampled real images (after grayscale conversion and resizing to $512 \times 512$). We then calculated the equivalent average spectrum for 100 images from each AI model and performed an element-wise subtraction (AI − Real). As shown, SD1.5 exhibits pervasive, diffuse high-frequency energy (red haze) which is indeed fragile to noise. In contrast, the SDXL model displays a distinct, structured geometric pattern (petal-like shape) with $3.0\times$ higher overall RMS difference. Paradoxically, this strong structural bias renders the generation artifacts *robust* rather than fragile to noise. Because RIGID assumes that "robustness" equates to "realness", it mistakenly classifies these stable SDXL samples as authentic. This demonstrates that even "model-agnostic" training-free methods are contingent on architectural assumptions that decay as generative models evolve, making a data-centric adaptation strategy imperative.

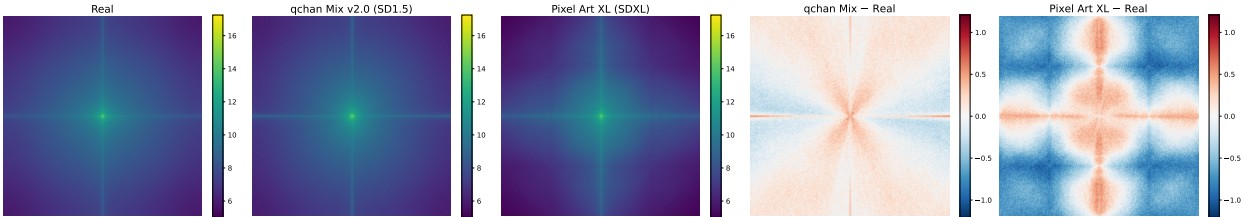

Figure 5: Spectral frequency analysis of residuals (AI − Real). The first three panels are the average log-magnitude FFT spectra of Real images, qchan Mix v2.0 (best detected by RIGID), and Pixel Art Diffusion XL (worst detected by RIGID). The last two panels are the corresponding residual maps: qchan Mix shows diffuse high-frequency artifacts (red haze), whereas Pixel Art Diffusion XL exhibits a distinct, structured geometric pattern (petal-like shape) with 3.0× higher RMS difference.

Table 4: Performance of training-free methods on the current WildGen validation split. For each of the 342 eligible AI sources, we pair its 100 validation images with a deterministic 100-image real subset drawn from the eight traceable real validation sources, then macro-average AP/AUC over sources.

| Method | AP (%) ↑ | AUC (%) ↑ |
|---|---|---|
| RIGID | 70.21 | 75.21 |
| AEROBLADE | 66.14 | 70.88 |
| WaRPAD | 57.14 | 61.32 |
| ManifoldBias | 42.76 | 37.13 |

## 5 Boundary Adaptation and Generalization

The zero-shot illusion documented in Section 4 raises a fundamental question: do detectors fail in the wild because current representations lack useful AI fingerprints, or because their decision boundaries are misaligned with the target distribution? In this section, we test that question with supervised boundary updates, source-identification probes, and external benchmark transfer. The results support a boundary-maintenance view: within WildGen, frozen features are highly separable and small updates are effective; across external benchmarks, the best adaptation strategy depends on the size and nature of the shift.

### 5.1 Evaluation Setup

To address the bottleneck of generalization, we investigate adaptation using two representative methods: CNNDetection (full fine-tuning of a pretrained ResNet-50 backbone) and UnivFD (linear probing on frozen CLIP ViT-L/14 features). We use a data-scaling protocol where each AI model contributes $N$ samples for training, with $N \in \{900, 90, 45, 18\}$.

Unless otherwise specified, the main split uses a real-to-AI ratio of 0.1: for each $N$, the number of real training images is 10% of the total generative samples. This ratio is a data-efficient compromise rather than a claim of optimality; Section 5.4 evaluates alternatives. We inversely scale training epochs $E$ with sample size $N$ so that each source contributes the same total number of training exposures (Table 6). Both detectors use batch size 64 and learning rate $10^{-4}$ (Adam for CNNDetection's full ResNet-50; AdamW on the frozen-CLIP linear head for UnivFD), with no early stopping and the final-epoch model reported. Evaluation is conducted on the fixed validation split: 34,200 AI images and 8,000 real images.

### 5.2 Binary Adaptation

Table 6 shows that both models reach high binary separation performance after adaptation. Since the validation split is imbalanced (34,200 AI vs. 8,000 Real images), we emphasize balanced accuracy and Real-class F1 rather than raw accuracy. UnivFD is particularly stable in low-data regimes, reaching 95.77% balanced accuracy at $N = 18$. CNNDetection benefits more from scale, improving from 85.25% at $N = 18$ to

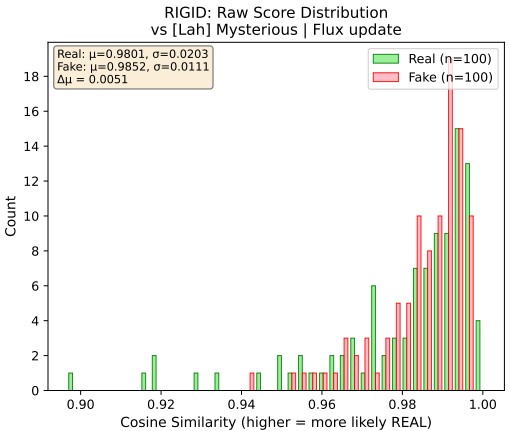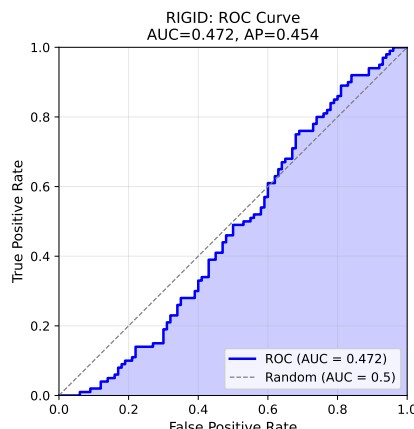

Figure 6: Visual analysis of training-free methods: (a) ROC curves and (b) score histograms.

Table 5: Top-3 and Bottom-3 performing models on WildGen based on AUC and AP scores for RIGID.

| Rank | Model Name | AUC ↑ | AP ↑ | Base Model |
|------|------------|-------|------|------------|
| *Best 3 Models* | | | | |
| 1 | qchan Mix v2.0 | 0.8839 | 0.8548 | SD 1.5 |
| 2 | Toon Babes v1.0 | 0.8456 | 0.8398 | SD 1.5 |
| 3 | AnyLoraCleanLinearMix-ClearVAE v1.0 | 0.8359 | 0.7736 | SD 1.5 |
| *Worst 3 Models* | | | | |
| 1 | Pixel Art Diffusion XL | 0.1795 | 0.3404 | SDXL |
| 2 | SVSN001 v1.0 | 0.3462 | 0.3937 | SDXL |
| 3 | ProtoVision XL - High Fidelity 3D | 0.3762 | 0.4072 | SDXL |

94.81% at $N = 900$. These results support the boundary-misalignment view: small amounts of target-domain data are sufficient to place a useful AI-vs-real boundary within WildGen.

The strong low-shot performance should not be read as universal cross-domain recovery: it reflects a shared train/validation feature geometry that a few examples can realign, whereas later cross-benchmark evaluations show that larger shifts may require more target-domain data or full fine-tuning.

## 5.3 Source Identification

Beyond binary detection, the 350-class identification task reveals the nuances of feature disentanglement. Unlike the binary task, real images are split into eight real-source classes, so Rec-R measures exact recovery of the real source rather than simply recognizing authenticity. As shown in Table 7, source identification is substantially harder than binary detection. CNNDetection achieves the strongest high-data source localization (53.94% top-1 and 78.46% top-5 at $N = 900$), while UnivFD preserves stronger real-source recall.

This complexity gap has two causes. First, the increase from 2 classes to 350 classes creates a crowded decision space. Second, many AI sources are phylogenetically related, often sharing base architectures and differing only by training data, fine-tuning, or adapters. We therefore treat source identification as forensic and diagnostic evidence rather than a solved exact-attribution task.

The optimization dynamics vary with the available data. In the data-rich regime ($N = 900$), CNNDetection's full parameter updates yield a Top-5 accuracy of 78.46%, which is forensically useful even when the exact checkpoint cannot be pinpointed. At $N = 18$, UnivFD instead has higher top-1 accuracy than CNNDetection (27.09% vs. 22.43%), reinforcing frozen features as a stable low-data representation. Thus $N = 18$ separates AI from Real in the binary task, but precise source characterization requires substantially more data.

Table 6: Binary adaptation metrics. We report final balanced accuracy and F1 for the Real class.

| $N$ | $E$ | UnivFD (Linear) | | CNNDet (Fine-tune) | |
|---|---|---|---|---|---|
| | | Balanced Acc. | F1(Real) | Balanced Acc. | F1(Real) |
| 900 | 10 | 97.22 | 96.79 | 94.81 | 93.71 |
| 90 | 100 | 96.97 | 96.45 | 93.77 | 90.83 |
| 45 | 200 | 96.69 | 96.11 | 91.10 | 88.54 |
| 18 | 500 | 95.77 | 94.94 | 85.25 | 80.99 |

Table 7: Source-identification metrics (%) on the 350-class source-multiclass split: 342 AI source classes plus 8 real-source classes. Rec-R is exact real-source recall.

| $N$ | $E$ | Top-1 (%) | Top-5 (%) | Rec-R (%) |
|---|---|---|---|---|
| | | *UnivFD (Linear Probe)* | | |
| 900 | 10 | 43.57 | 63.85 | 82.78 |
| 90 | 100 | 37.17 | 56.59 | 78.28 |
| 45 | 200 | 32.61 | 51.44 | 75.38 |
| 18 | 500 | 27.09 | 44.51 | 70.61 |
| | | *CNNDetection (Fine-tune)* | | |
| 900 | 10 | 53.94 | 78.46 | 72.14 |
| 90 | 100 | 33.74 | 56.62 | 60.03 |
| 45 | 200 | 26.59 | 46.27 | 53.94 |
| 18 | 500 | 22.43 | 40.49 | 53.78 |

## 5.4 Real-Source Bias and Real-Anchor Ratio

The preceding binary results pool all real images into one class, but the real distribution itself is heterogeneous. To quantify this effect, we train an 8-way classifier on frozen CLIP features to predict the real-image source. It reaches 84.33% balanced accuracy, confirming that real-source identity is highly learnable (per-source precision/recall in Table 10). This does not invalidate binary detection, but it means that Real cannot be treated as a homogeneous background distribution. This compounds the narrow-Real benchmark problem discussed in Section 3: when the fake generator varies but the Real class stays fixed (e.g., LSUN or ImageNet), a detector can learn the benchmark's specific Real distribution rather than a general notion of authenticity. The source-specific FPR columns in Table 3 test this risk directly by evaluating pretrained legacy detectors on the eight real validation sources in WildGen.

Table 3 provides direct evidence that real-side shortcuts contribute to the zero-shot illusion. The effect is not uniform across methods: AlignedForensics, CNNDetection, UnivFD, and RINE remain conservative on these real sources, while DIRE, NPR, AIDE, Effort, and LGrad misclassify a large fraction of specific real domains as fake. The concentration on YFCC/Flickr CC, Unsplash, and SUN397 suggests that detectors trained on narrow real distributions can treat unfamiliar real-world photography as out-of-distribution fake content. We therefore treat real-source diversity as a core benchmark requirement rather than a cosmetic dataset expansion.

Table 8: Real-to-AI ratio sensitivity. Balanced accuracy (%) is reported for binary pooled-real training.

| Model | Real Ratio | $N = 18$ | $N = 900$ |
|---|---|---|---|
| UnivFD | 0.01 | 85.10 | 91.20 |
| UnivFD | 0.10 | 95.77 | 97.22 |
| UnivFD | 1.00 | 98.61 | 98.88 |
| CNNDet | 0.01 | 62.70 | 86.67 |
| CNNDet | 0.10 | 85.25 | 94.81 |
| CNNDet | 1.00 | 95.95 | 98.64 |

Table 8 shows why the main protocol uses a 0.1 real-to-AI ratio: it provides a useful real anchor at much lower data cost than fully balanced training. However, lowering the real ratio to 0.01 substantially harms balanced accuracy, especially for CNNDetection at $N = 18$. We therefore report real-source bias and ratio sensitivity as controls, and avoid interpreting the Real class as a monolithic concept.

Table 9: Cross-benchmark joint few-shot adaptation (balanced accuracy, %). At $K$=900 the WildGen side is the full training pool, i.e. a full-data reference rather than a few-shot point. The "WildGen" column re-evaluates each $K$=900 adapted model on the WildGen validation split to measure forgetting.

| | | Target balanced acc. | | | | WildGen |
|---|---|---|---|---|---|---|
| Model | Target | Zero-shot | $K$=18 | $K$=90 | $K$=900 | ($K$=900) |
| UnivFD | GenImage | 61.11 | 80.01 | 83.92 | 85.02 | 96.31 |
| UnivFD | ForenSynths | 51.32 | 85.68 | 89.24 | 90.05 | 96.90 |
| CNNDet | GenImage | 60.68 | 72.43 | 79.11 | 85.94 | 94.94 |
| CNNDet | ForenSynths | 51.47 | 71.91 | 80.49 | 94.20 | 95.55 |

## 5.5 Cross-Benchmark Transfer

Section 4 showed detectors trained on existing benchmarks fail on WildGen; we now test the converse. We adapt both families to GenImage and ForenSynths via *balanced joint few-shot fine-tuning*: warm-starting from the WildGen detector and fine-tuning on $K$ images per source drawn from both WildGen and the target, with the target real anchor aligned to the real pool.

As shown in Table 9, both families recover from near-chance zero-shot with only a few labelled examples per source. UnivFD is the most data-efficient—at $K$=18 it already reaches 80.0 (GenImage) and 85.7 (ForenSynths)—while CNNDetection needs more target data but scales higher on the most distant benchmark, overtaking UnivFD on ForenSynths at $K$=900 (94.2 vs. 90.1). The best strategy thus depends on target-domain distance and trainable capacity, not a single recipe.

Two controls shape these results. *Real-source anchoring is decisive*: under the default 0.1 real ratio the target's own real images are a negligible fraction of the mix and both detectors over-reject them as fake (e.g. UnivFD ForenSynths real recall 57%), capping balanced accuracy at 76–86%; aligning the target real count restores real recall to 91–94% and gives the numbers above. This is the real-source bias of Section 5.4 acting on the AI-vs-real boundary. *Adaptation maintains rather than replaces the boundary*: re-evaluated on WildGen, adapted models still score $\geq 96\%$ (UnivFD) and $\geq 94\%$ (CNNDetection) balanced accuracy at $K$=900. Few-shot replay therefore *extends* the valid boundary to a new benchmark while preserving the old one.

## 5.6 Environmental Robustness

Finally, we evaluate JPEG compression, resizing, and their combination at $N = 18$ and $N = 900$. These perturbations are not the primary failure mode in our evaluation. For UnivFD, JPEG compression causes the largest drop—balanced accuracy falls from 96.0% to 93.9% at $N = 18$ and from 97.2% to 95.2% at $N = 900$ (about 2 points)—while resizing alone leaves it essentially unchanged (within 0.6 points). CNNDetection shows no meaningful degradation: its balanced accuracy stays at or slightly above the clean baseline under every perturbation (90.7% clean vs. 91.3–91.8% at $N = 18$; 96.4% clean vs. 96.4–96.5% at $N = 900$). We therefore treat environmental robustness as supporting evidence: distribution shift and real-source confounding remain the more important failure modes.

## 6 Conclusion

We expose the zero-shot illusion in AIGC detection: detectors that appear to generalize on curated benchmarks fail sharply in the wild setting of WildGen. This failure is one of boundary misalignment rather than missing features—frozen CLIP features remain highly separable within WildGen—but two factors complicate it: narrow Real distributions act as shortcuts, producing real-source bias and source-dependent false positives, and transfer to external benchmarks such as GenImage and ForenSynths exposes large distribution shifts. Few-shot adaptation is thus a maintenance mechanism, not a universal guarantee: frozen-feature probes suffice for same-domain updates, full fine-tuning absorbs larger cross-benchmark shifts, and both need real-source controls. We therefore argue for reporting the distributional boundary within which a detector is valid, and hope WildGen offers a reproducible foundation as generators and real-image sources evolve.

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

## A    Generated and Real Samples

In this appendix, we provide additional visual examples to illustrate the diversity of the WildGen. Figure 7 showcases images generated using complex synthesis pipelines, including high-fidelity models like SDXL and FLUX.dev, along with their detailed prompting and configuration metadata. Figure 8 presents examples of real images from our dataset, which serve as the authentic baseline for our binary classification and source-identification evaluations.

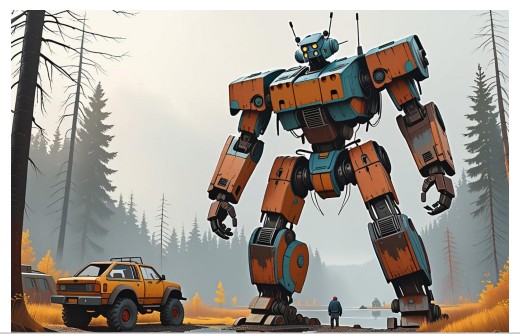
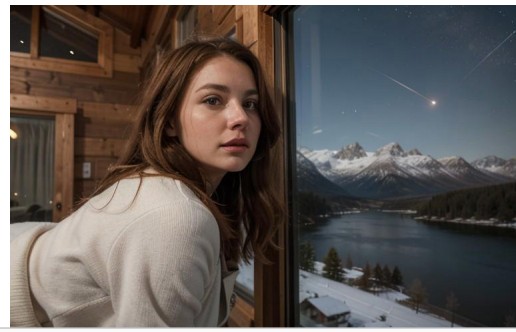

| Pipe: | [Base] AAM XL (Anime Mix) · [LoRA] Perfect Eyes XL:0.65, Envy Mimic XL 01:0.35, extremely detailed (no trigger):0.65, Red Grunge:0.35, Midjourney mimic:0.65, Simon Stalenhag TFTL SDXL:0.45, Pink Pearl:0.35 · [Embeds] Ultimate Text Embeddings SDXL Pack |
|---|---|
| Prompt: | Cinematic.  Dynamic.  Ancient, rusty and tall mech warrior.  Cut away diagram.  Many details everywhere...  [truncated] |
| Neg: | 35mm film, deformed, glitch, noisy, signature, watermark.  (worst quality, low quality:1.4), (easynegative, negative_hand-neg), (crown:1.5), horns, long ears, stars. |
| Cfg: | Steps:  50 · CFG: 7.0 · Sampler:  Euler A |

| Pipe: | [Base] epiCRealism |
|---|---|
| Prompt: | score_9, score_8_up, score_7_up, realistic lighting, photo, photorealistic Blonde red hair 30 year old woman with green eyes, leaning out a window looking at the star field night sky.  A lake, valley, cabin porch...  [truncated] |
| Neg: | score_6, score_5, score_4, source_pony, source_anime, source_furry, source_cartoon, (worst quality:2), (low quality:2), (normal quality:2), (lowres, error), username, watermark, ugly face |
| Cfg: | Steps:  50 · CFG: 4.0 · Sampler:  DPM++ 2M Karras |

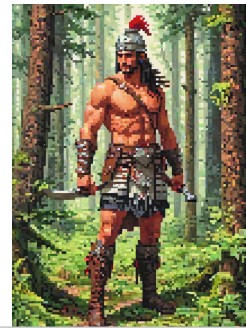
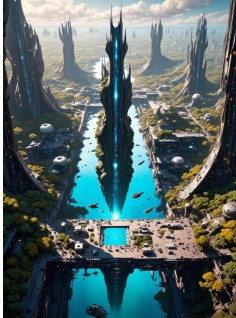

| Pipe: | [Base] Pixel Art Diffusion XL |
|---|---|
| Prompt: | Gaulish warrior, full body, 16bit, in european forest |
| Neg: | bad quality, bad anatomy, worst quality, low quality, low resolution, extra fingers, blur, blurry, ugly, wrong proportions, watermark, image artifacts, lowres, ugly, jpeg artifacts, deformed, noisy image |
| Cfg: | Steps:  20 · CFG: 3.0 · Sampler:  DPM++ 2M |

| Pipe: | [Base] FLUX dev · [LoRA] Velvet's Mythic Fantasy Styles:1.0, Sci-fi Environments:0.85, Artify's Fantastic Flux Landscape Lora:1.0 |
|---|---|
| Prompt: | bustling sci-fi metropolis, chrome towers, a bridge running over a clear, fast-flowing large river with azure waters, mythp0rt, sunset, vivid vibrant colours, yellow vibrant trees, flying vehicles |
| Cfg: | Steps:  25 · CFG: 3.5 |

Figure 7: Examples of complex generation pipelines in the WildGen. Each example highlights the detailed configuration and prompting strategies used to synthesize high-quality images.

## B    Qualitative LMM Case Study

This appendix presents a qualitative case study of Large Multimodal Models (LMMs) for AI-image detection. It is not a systematic benchmark and should be read only as an illustration of how general-purpose assistants reason about provenance. We queried Gemini and ChatGPT in January 2026 using a sample image released alongside Z-Image (Z-Image Team, 2025) in December 2025 (Figure 9).

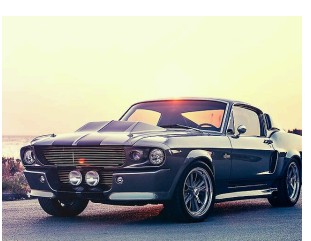 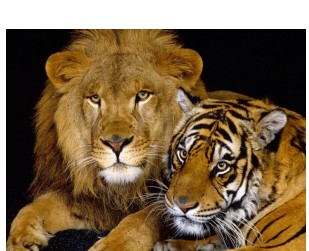 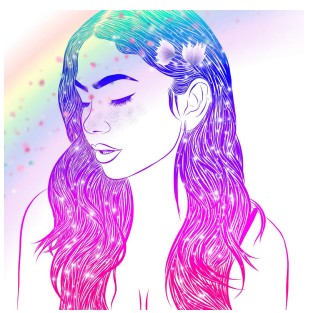 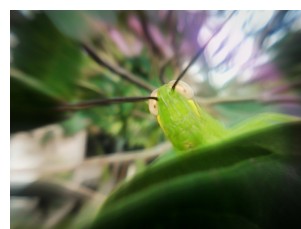

Figure 8: Examples of real (authentic) images in WildGen.

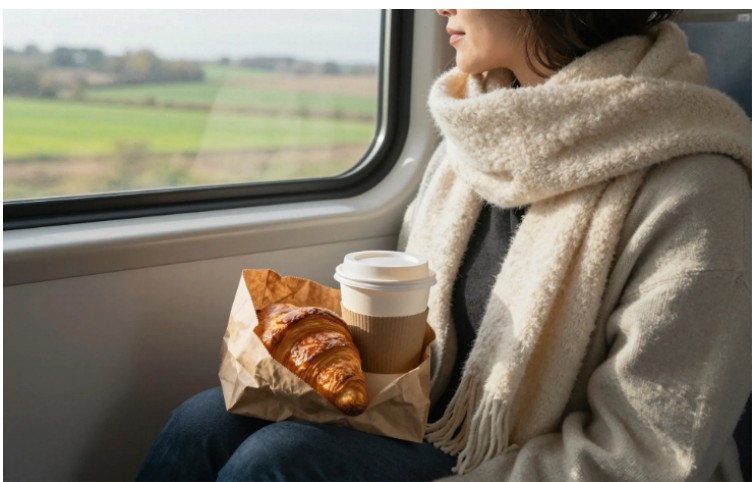

Figure 9: The sample image from Z-Image (December 2025) used to evaluate LMM detection capabilities.

### B.1 Gemini Evaluation

We first queried Gemini. Its default response, which relies partly on digital watermark detection, was as follows:

> *Based on a digital watermark check, this image was not made with Google's AI tools. While it doesn't contain a Google AI watermark, it's difficult to say for certain if it was created using other AI platforms or if it's a real photograph. The lighting and textures, such as the knit of the scarf and the glaze on the croissant, look very natural. However, many modern AI generators are capable of producing highly realistic results.*

When explicitly instructed not to rely on built-in watermark detection tools, Gemini provided the following analysis:

> *It's difficult to say for certain if an image is AI-generated just by looking at it, but there are a few common signs you can check for:*
> ***Inconsistencies in Detail:*** *AI often struggles with complex details. Look closely at things like hands, eyes, and hair. You might see extra fingers, asymmetrical features, or hair that seems to blend into the skin.*
> ***Strange Textures:*** *Some AI images have a "too perfect" or plastic-like texture, while others may have blurry or smeared areas in the background that don't make sense.*
> ***Logical Errors:*** *Look for things that defy physics or logic, like objects melting into each other, shadows going in different directions, or text that is nonsensical or uses unrecognizable*

*characters.*

***Metadata and Watermarks:*** *Sometimes the image file itself will have metadata indicating it was created with an AI tool, or there might be a small watermark from a service like DALL-E or Midjourney.*

*If you have a high-resolution version of the image, these details are often easier to spot. You can also try using online AI detection tools, though they are not always 100% accurate.*

## B.2 ChatGPT Evaluation

We also queried ChatGPT, which provided a detailed visual inspection:

*I can't be 100% certain, but based on visual inspection alone, this image looks more like a real photograph than an AI-generated one.*

*Here's why:*

***Natural lighting and reflections:*** *The light coming through the train window, the soft shadows, and the reflections on the glass look physically consistent and subtle in a way that current AI images often struggle with.*

***Material realism:*** *The textures of the scarf, paper bag, croissant layers, and coffee cup sleeve appear coherent and physically plausible, without the smeared or overly perfect details often seen in AI outputs.*

***Composition and imperfections:*** *The framing is slightly off-center and casual, like a candid travel photo, rather than overly stylized or symmetric.*

***No obvious AI artifacts:*** *I don't see common signs such as warped objects, inconsistent edges, strange text, or anatomically odd details.*

*That said, modern AI models can produce images of this quality, especially for lifestyle scenes like this. Without metadata (EXIF), provenance, or known source context, the safest conclusion is:*

***Likely a real photo, but high-quality AI generation cannot be ruled out.***

*If you want, I can: Point out specific AI telltale signs to double-check Analyze it assuming a particular model family (e.g., diffusion vs. GAN) Help assess likelihood using context such as where the image came from*

## B.3 Analysis and Implications

These examples suggest that LMMs often reason through visible artifacts, metadata, and watermark availability, rather than applying a calibrated detector. This behavior can be useful for qualitative triage, but it is not a substitute for a controlled detection benchmark.

The case study is consistent with the main paper's broader message: provenance judgments depend on distributional context and available evidence. However, because this appendix contains only a small number of prompts and model responses, we do not use it as quantitative evidence for or against LMM-based detection. A systematic LMM evaluation would require fixed model versions, repeated prompts, a balanced image set, and a predefined scoring protocol.

## C   Real-Source Separability Breakdown

This appendix expands the real-source bias control in Section 5.4. We train a logistic-regression classifier on frozen CLIP ViT-L/14 features to predict which of the eight real-image sources an authentic image comes from, using 5,000 training and 1,000 validation images per source (seed 42). The classifier reaches 84.33% balanced accuracy; Table 10 reports the per-source precision, recall, and F1.

The sources are far from equally separable. Curated, visually distinctive collections such as Food101 and Unsplash are almost perfectly recovered (99.50% and 95.50% recall), whereas broad web-photo sources overlap

more: OpenImages (67.10% recall) and SUN397 (72.60%) are the hardest to distinguish. This confirms that "Real" is not a homogeneous background class, and that a detector trained on any single real source can inherit that source's idiosyncrasies rather than a general notion of authenticity.

Table 10: Per-source performance of the frozen-CLIP eight-way real-source classifier on the WildGen real validation split (1,000 images per source). Overall balanced accuracy is 84.33%.

| Real Source | Precision (%) | Recall (%) | F1 (%) |
|---|---|---|---|
| Food101 | 96.88 | 99.50 | 98.17 |
| Unsplash | 91.30 | 95.50 | 93.35 |
| Places365 | 89.61 | 92.30 | 90.94 |
| Google Landmarks v2 | 83.69 | 86.20 | 84.93 |
| ADE20K | 81.36 | 81.60 | 81.48 |
| YFCC/Flickr CC | 83.21 | 79.80 | 81.47 |
| SUN397 | 74.46 | 72.60 | 73.52 |
| OpenImages | 72.15 | 67.10 | 69.53 |
| Macro average | 84.08 | 84.33 | 84.17 |

## D  Generative Model Catalog

This appendix lists the 342 AI sources included in the main WildGen benchmark split. Each source has at least 1,000 indexed images; the split protocol selects 1,000 images per source, with 900 images assigned to the train pool and 100 images assigned to validation. Model IDs 438–447 correspond to recently released generators and are included in this default split on the same terms as all other sources.

Table 11: AI sources with $\geq 1,000$ indexed images included in the main WildGen split.

| Model ID | Source Name | Indexed Images |
|---|---|---|
| 1 | +RealisticAnimeHentMix666+v1.0 | 1,415 |
| 2 | 287641 | 1,026 |
| 3 | 2DN-Pony | 3,218 |
| 4 | 2DN-Ponyv1 | 1,854 |
| 6 | 595775 | 1,188 |
| 7 | 667593 | 1,166 |
| 8 | AAA(AAAAAAAAAAAAAAAAAAAA) \| Finetune mix on whatever model i want at that point which is Illustrious XL right now,but i will keep Pony one for on-site as well | 3,498 |
| 10 | AAM XL (Anime Mix) | 4,337 |
| 11 | AAM XL (Anime Mix)v1.0 | 3,480 |
| 12 | AbsoluteReality | 4,008 |
| 14 | AbsoluteRealityv1.8.1 | 4,660 |
| 15 | AbyssOrangeMix2 - Hardcore | 1,248 |
| 16 | AbyssOrangeMix2 - HardcoreAbyssOrangeMix2_hard | 2,140 |
| 17 | AlbedoBase XL | 4,718 |
| 18 | AlbedoBase XLv1.3 | 2,097 |
| 19 | AlbedoBase XLv2.0 | 3,339 |
| 20 | AlbedoBase XLv2.1 | 4,112 |
| 22 | Analog Madness - Realistic model | 3,196 |
| 24 | Analog Madness - Realistic modelv7.0 | 3,914 |
| 26 | AniMerge | 1,964 |
| 29 | AniMesh | 1,492 |
| 30 | AniMeshAnimesh-Pruned V2.2 | 1,776 |
| 32 | AniVerse | 3,797 |
| 33 | AniVerseV1.5 - Pruned | 1,628 |
| 34 | AniVerseV2.0 (HD) - Pruned | 1,367 |

Table 11, continued from previous page

| Model ID | Source Name | Indexed Images |
|---|---|---|
| 36 | Animagine XL V3.1 | 4,681 |
| 37 | Animagine XL V3.1v3.0 | 4,740 |
| 38 | Animagine XL V3.1v3.1 | 4,689 |
| 39 | Anime Illust Diffusion XL | 1,723 |
| 41 | Anime Model OtakuReliableEnable (AMORE) | 2,378 |
| 42 | AnyLoRA - Checkpoint | 3,694 |
| 43 | AnyLoRA - CheckpointbakedVae (blessed) fp16 NOT-PRUNED | 4,565 |
| 44 | AnyLoRA - CheckpointbakedVae ftMse fp16 NOT-PRUNED | 3,496 |
| 45 | AnyLoraCleanLinearMix-ClearVAE | 4,028 |
| 46 | AnyLoraCleanLinearMix-ClearVAEv1.0 | 4,883 |
| 47 | AnyOrangeMix - Anything + AbyssOrangeMix | 1,962 |
| 48 | AnythingElse V4v4.5 | 2,383 |
| 49 | Arthemy Comics | 2,246 |
| 51 | Artiumv2.0 | 1,304 |
| 53 | AutismMix SDXL | 4,739 |
| 55 | AutismMix SDXLAutismMix_confetti | 4,919 |
| 56 | AutismMix SDXLAutismMix_pony | 4,844 |
| 57 | BB95 Furry Mix | 3,496 |
| 58 | Beautiful Realistic Asiansv7 | 1,042 |
| 59 | Better than words | 3,243 |
| 60 | Better than wordsv3.0 | 1,232 |
| 61 | BlenderMix | 1,744 |
| 62 | Boltning - Realistic, Lightning, HYPERHYPER_D | 1,030 |
| 64 | BriXL | A must in your toolbox | 1,562 |
| 66 | CATalyst - Citron Anime Treasure Catalystv1.0 | 1,032 |
| 67 | CG Realistic PonyV5 | 3,292 |
| 68 | CalicoMix | 3,353 |
| 71 | CashMoney (Anime) | 1,803 |
| 72 | Cherry Picker XL | 1,592 |
| 73 | ChilloutMixChilloutmix-Ni-pruned-fp32-fix | 4,249 |
| 74 | CineVisionXL by SoCalGuitaristRelease-v1.5.0-BakedVAE | 1,292 |
| 75 | Cinenauts XL (A TRUE CINEMATIC DIFFUSION) | 1,957 |
| 76 | Cinenauts XL (A TRUE CINEMATIC DIFFUSION)Cinenauts v1.0 | 3,421 |
| 77 | Cinenauts XL (A TRUE CINEMATIC DIFFUSION)Cinenauts v2.0 | 2,001 |
| 78 | CoffeeBreakv2.0 | 1,426 |
| 79 | Copax TimeLessXL | 3,517 |
| 80 | Copax TimeLessXLV8 | 2,319 |
| 83 | Counterfeit-V3.0 | 3,811 |
| 84 | Counterfeit-V3.0v3.0 | 4,258 |
| 85 | Crystal Clear One_vS1 | 1,434 |
| 86 | Crystal Clear XL | 3,926 |
| 87 | Crystal Clear XL - Primev1.0 | 1,852 |
| 88 | Crystal Clear XLCCXL | 4,442 |
| 89 | CyberRealistic | 4,402 |
| 90 | CyberRealistic Pony | 3,191 |
| 91 | CyberRealistic Ponyv6.3 | 1,114 |
| 92 | CyberRealisticv3.3 | 3,328 |
| 94 | CyberRealisticv5.0 | 4,191 |
| 95 | CyberSushi | 1,798 |
| 96 | CyberSushiv2.1 | 1,040 |
| 97 | DENRealMergev1.0 | 4,019 |
| 98 | Dark Sushi 2.5D 2.5D | 1,851 |
| 99 | Dark Sushi 2.5D 2.5Dv4.0 | 1,201 |
| 100 | Dark Sushi Mix Mix | 3,762 |
| 101 | Dark Sushi Mix Mix2.25D | 3,545 |

Table 11, continued from previous page

| Model ID | Source Name | Indexed Images |
|---|---|---|
| 102 | DarkSunv4.1 | 1,690 |
| 103 | DevlishPhotoRealism SDXL | 2,009 |
| 105 | DisneyRealCartoonMix | 2,245 |
| 106 | Divine Diffusion by Dever (DDD) [SDXL] | 2,514 |
| 109 | Doll-E XLv1 | 1,721 |
| 110 | DreamShaper | 4,832 |
| 111 | DreamShaper XL | 4,660 |
| 112 | DreamShaper XLLightning DPM++ SDE | 2,097 |
| 114 | DreamShaper XLalpha2 (xl1.0) | 4,698 |
| 115 | DreamShaper XLv2.1 Turbo DPM++ SDE | 2,873 |
| 117 | DreamShaper8 | 3,854 |
| 118 | DucHaiten-AIart-SDXL | 3,312 |
| 119 | DucHaiten-AIart-SDXLv2.0 | 1,119 |
| 120 | DucHaiten-AIart-SDXLv3.3.5.1.5 | 1,167 |
| 121 | DucHaiten-Pony-Real | 3,805 |
| 122 | DucHaiten-Pony-XL (no-score) | 4,613 |
| 124 | DucHaiten-Pony-XL (no-score)pony-no-score_v4.0 | 2,831 |
| 126 | DucHaiten-Pony-XL (no-score)v5.2 | 1,456 |
| 129 | DynaVision XL - All-in-one stylized 3D SFW and NSFW output, no refiner needed! | 4,032 |
| 130 | DynaVision XL - All-in-one stylized 3D SFW and NSFW output, no refiner needed!Release_0.5.5.7.BakedVAE | 2,082 |
| 131 | DynaVision XL - All-in-one stylized 3D SFW and NSFW output, no refiner needed!Release_v0.6.1.0-bakedvae | 1,589 |
| 136 | Ether PDXL | 2,006 |
| 137 | Everclear PNY by Zovya | 3,170 |
| 138 | FLUX | 4,886 |
| 139 | FLUX.1 [dev]v1.0 | 2,645 |
| 140 | FLUX.1DEV FP8 - Kijai [11 GB] | 1,683 |
| 141 | Fantasm | Pony XL | 1,453 |
| 142 | FenrisXL + Flux | 3,493 |
| 143 | Flat-2D Animerge | 1,692 |
| 144 | ForArtisticXLv0.1 | 1,342 |
| 145 | ForRealXL V1.0v0.1 | 2,208 |
| 146 | ForRealXL V1.0v0.3 | 1,080 |
| 147 | ForRealXL V1.0v0.5 | 3,880 |
| 151 | FurryBlend | 3,130 |
| 156 | GhostMix | 3,418 |
| 157 | GhostMixv2.0-BakedVAE | 4,414 |
| 158 | Hassaku (hentai model) | 4,511 |
| 159 | Hassaku (hentai model)v1.3 | 4,088 |
| 160 | Hassaku XL (Hentai) | 4,156 |
| 161 | Hassaku XL (Hentai)v1.3 | 3,075 |
| 162 | Hassaku XL (Hentai)v1.3(better eyes version) | 1,437 |
| 166 | Honey Mix [XL] High contrast anime checkpoint | 1,190 |
| 167 | Hoseki - LustrousMix [Pony XL] | 2,792 |
| 168 | Hoseki - LustrousMix [Pony XL]v1.0 | 1,743 |
| 170 | ICBINP - "I Can't Believe It's Not Photography"Final | 1,147 |
| 172 | Incursio's Meme Diffusion (SDXL, Pony) | 4,542 |
| 173 | Incursio's Meme Diffusion (SDXL, Pony)v1.6PDXL | 4,476 |
| 174 | Incursio's Meme Diffusionv1.6 Pruned | 1,022 |
| 175 | Indigo Furry mix | 4,074 |
| 180 | Jib Mix Realistic XL | 4,211 |
| 181 | Jib Mix Realistic XLv9.0 Better Bodies | 1,933 |
| 182 | Juggernaut | 3,323 |

Table 11, continued from previous page

| Model ID | Source Name | Indexed Images |
|---|---|---|
| 183 | Juggernaut XL | 4,882 |
| 184 | Juggernaut XLJugg_XI_by_RunDiffusion | 1,051 |
| 185 | Juggernaut XLJugg_X_by RunDiffusion | 4,026 |
| 186 | Juggernaut XLV 7 + RunDiffusion | 4,317 |
| 187 | Juggernaut XLV 8 + RunDiffusion | 4,300 |
| 188 | Juggernaut XLV9 + RunDiffusionPhoto 2 | 4,188 |
| 189 | Juggernaut XLV9+RDPhoto2-Lightning_4S | 3,261 |
| 190 | Juggernaut XLVersion 6 + RunDiffusion | 4,728 |
| 191 | JuggernautReborn | 2,326 |
| 192 | Kohaku-XL beta | 1,444 |
| 193 | Kohaku-XL betabeta7 | 2,399 |
| 194 | Koji | 3,305 |
| 195 | Kojiv2.1 | 1,891 |
| 196 | LEOSAM's HelloWorld XL | 4,001 |
| 197 | LEOSAM's HelloWorld XLHelloWorld XL 5.0 GPT4V | 1,881 |
| 198 | LEOSAM's HelloWorld XLHelloWorld XL 6.0 | 1,346 |
| 199 | LEOSAM's HelloWorld XLHelloWorld XL 7.0 | 2,932 |
| 201 | LazyMix+ (Real Amateur Nudes) | 1,146 |
| 202 | Luminaverse [Pony XL] | 4,177 |
| 203 | Luminaverse [Pony XL]v1.0 | 1,253 |
| 206 | MFCG PDXL | 2,179 |
| 207 | MFCG PDXLv1.0 | 2,617 |
| 208 | MHXL - Aventis Horizon (SDXL) | 1,307 |
| 209 | MeichiDarkMix_Reload | 1,955 |
| 210 | MeinaHentai | 3,664 |
| 211 | MeinaHentaiV4 | 2,843 |
| 212 | MeinaMix | 4,325 |
| 213 | MeinaMixMeina V11 | 4,851 |
| 214 | MeinaPastel | 2,365 |
| 215 | MeinaPastelV6 ( Pastel ) | 1,907 |
| 216 | MeinaUnreal | 1,017 |
| 217 | MeinaUnrealV4.1 | 2,218 |
| 218 | Mistoon_Anime | 3,571 |
| 220 | Moxie Diffusion XLv1.32 | 1,010 |
| 223 | NeverEnding Dream (NED) | 2,535 |
| 225 | NightVisionXL | 4,578 |
| 226 | NightVisionXLNightVisionXL_v9.0.0 | 1,902 |
| 227 | NightVisionXLRelease_0.7.7.0.BakedVAE | 1,538 |
| 228 | NightVisionXLV0.7.9.1.BakedVAE | 2,896 |
| 229 | NightVisionXLv0.8.1.1 | 1,706 |
| 231 | ONE FOR ALL Pony Fantasy DPO+VAE | 2,340 |
| 233 | Objective Reality (Reloaded)v2.0 (original VAE) | 2,802 |
| 235 | Omnium | 2,472 |
| 236 | Omniumv1.1 | 1,080 |
| 238 | Painter's Checkpoint (oil paint / oil painting art style) | 1,252 |
| 239 | Paradox 2 SD XL 1.0Paradox 2 SD XL 1.0 | 1,082 |
| 241 | Perfect World | 3,267 |
| 243 | PerfectDeliberate | 1,422 |
| 244 | PhotoPedia XL | 2,962 |
| 245 | PhotoPedia XL4.5 | 3,183 |
| 246 | Photon | 1,489 |
| 247 | Photonv1 | 1,843 |
| 248 | PicX_real | 3,433 |
| 249 | PicX_real1.0 | 3,981 |
| 250 | Pixel Art Diffusion XL | 1,645 |

Table 11, continued from previous page

| Model ID | Source Name | Indexed Images |
|---:|---|---:|
| 251 | Pony Diffusion V6 XL | 4,944 |
| 252 | Pony Diffusion V6 XLV6 (start with this one) | 4,652 |
| 254 | Pony Realism | 4,490 |
| 255 | Pony Realism v2.1 Main + VAE | 3,943 |
| 256 | Pony FaeTality | 4,372 |
| 257 | Pony FaeTalityv1.0 | 2,088 |
| 258 | Pony FaeTalityv1.1 | 3,680 |
| 260 | Prefect Pony XL | 4,569 |
| 261 | Prefect Pony XLv1.0 | 2,687 |
| 262 | Prefect Pony XLv2 cleaned style | 2,622 |
| 263 | Prefect Pony XLv3 | 1,411 |
| 265 | ProtoVision XL - High Fidelity 3D / Photorealism / Anime / hyperrealism - No Refiner Needed | 4,307 |
| 266 | ProtoVision XL - High Fidelity 3D / Photorealism / Anime / hyperrealism - No Refiner NeededRelease_0.6.3.0.BakedVAE | 1,172 |
| 267 | ProtoVision XL - High Fidelity 3D / Photorealism / Anime / hyperrealism - No Refiner NeededRelease_v6.6.0-BakedVAE | 1,917 |
| 270 | RPG | 1,630 |
| 272 | Raemora XLv1.0 | 1,134 |
| 274 | ReV Animated | 4,250 |
| 275 | ReV AnimatedV2 Rebirth | 1,164 |
| 276 | ReV Animatedv1.2.2-EOL | 4,804 |
| 277 | Real Dream | 3,522 |
| 278 | RealCartoon-Pixar | 1,829 |
| 279 | RealCartoon-Realistic | 3,152 |
| 280 | RealCartoon-XL | 4,557 |
| 281 | RealCartoon-XLV5 | 1,078 |
| 282 | RealCartoon-XLV6 | 3,454 |
| 283 | RealCartoon3D | 3,634 |
| 284 | RealCartoon3DV15 | 1,126 |
| 285 | RealVisXL V5.0 | 4,615 |
| 286 | RealVisXL V5.0V2.0 (BakedVAE) | 2,321 |
| 287 | RealVisXL V5.0V3.0 (BakedVAE) | 2,300 |
| 288 | RealVisXL V5.0V4.0 (BakedVAE) | 4,164 |
| 290 | Realisian | 3,057 |
| 291 | Realism Engine SDXL | 3,274 |
| 292 | Realism Engine SDXLv3.0 VAE | 2,634 |
| 293 | Realistic Stock Photov2.0 | 1,200 |
| 294 | Realistic Vision V6.0 B1 | 4,303 |
| 295 | Realistic Vision V6.0 B1V5.1 (VAE) | 4,783 |
| 296 | Realistic Vision V6.0 B1V5.1 Hyper (VAE) | 1,512 |
| 299 | Rev Engine PonyXLv1.0 | 1,523 |
| 300 | SD XL | 4,842 |
| 301 | SD XLv1.0 | 4,907 |
| 302 | SD XLv1.0 VAE fix | 4,481 |
| 303 | SDVN6-RealXL | 1,903 |
| 304 | SDVN6-RealXLDetailface | 2,047 |
| 305 | SDVN7-NijiStyleXL | 3,829 |
| 306 | SDVN7-NijiStyleXLv1 | 1,347 |
| 308 | SDXL Unstable Diffusers YamerMIX | 4,691 |
| 309 | SDXL Unstable Diffusers YamerMIXNihilMania | 1,762 |
| 310 | SDXL Unstable Diffusers YamerMIXV11 + RunDiffusion | 2,841 |
| 312 | SDXL Unstable Diffusers YamerMIXV8 Heaven's Wrath+VAE | 4,039 |
| 313 | SDXL Yamer's Anime Unstable Illustrator | 2,902 |
| 314 | SDXL Yamer's Realism! - Realistic/Anime/3D | 1,896 |

Table 11, continued from previous page

| Model ID | Source Name | Indexed Images |
|---:|---|---:|
| 315 | SDXL Yamer's Realistic 5 | 4,400 |
| 316 | SDXL Yamer's Realistic 5 V5 + RunDiffusion | 1,854 |
| 317 | SDXL_Niji_Seven | 4,174 |
| 318 | SDXL_Niji_SevenSDXL_Niji_SE | 1,980 |
| 319 | SDXL_Niji_SevenSDXL_Niji_Seven | 1,048 |
| 320 | SDXL_Niji_SevenSDXL_Niji_v6 | 1,220 |
| 321 | SDXXXL | 1,582 |
| 322 | SVSN001v1.0 | 4,775 |
| 323 | SereneXL | 1,605 |
| 324 | SkibidiMix | 1,777 |
| 325 | SoloMixv1.0 | 4,532 |
| 326 | Starlight XL Animated | 4,507 |
| 327 | Starlight XL Animatedv3 | 3,771 |
| 328 | Sym-Pony World | 2,818 |
| 329 | Sym-Pony Worldv1.0 | 1,109 |
| 330 | T-ponynai3 | 2,278 |
| 332 | T-ponynai3v6.5 | 1,028 |
| 334 | TalmendoXL - SDXL Uncensored Full Model | 1,148 |
| 335 | Tamarin_XLv1.0 | 3,889 |
| 336 | The Truality Engine | 1,270 |
| 338 | The Truality EngineThe Truality Engine V3 | 1,680 |
| 340 | Toon Babes | 1,419 |
| 341 | Toon Babesv1.0 | 4,151 |
| 342 | ToonYou | 2,497 |
| 344 | Toonify | 3,060 |
| 348 | Unleashed DiffusionWhimsical | 1,717 |
| 349 | Virtual Diffusion Pony XL | 1,616 |
| 350 | WAI-ANI-NSFW-PONYXL | 3,600 |
| 351 | WAI-ANI-NSFW-PONYXLv7.0 | 2,182 |
| 352 | WAI-ANI-NSFW-PONYXLv8.0 | 1,394 |
| 355 | WildCardX-XL LIGHTNINGWildCardX-XL+Rundiffusion | 2,603 |
| 356 | WildCardX-XL-Fusion | 4,731 |
| 358 | WildCardX-XL-FusionFUSION OG | 3,347 |
| 359 | WildCardX-XLV4+RunDiffusion | 3,096 |
| 360 | WoW_(XL+PD+Flux). | 1,517 |
| 361 | WoW_(XL+PD+Flux).v2 | 3,851 |
| 362 | XL6 - HEPHAISTOS SD 1.0XL (SFW&NSFW) | 3,052 |
| 363 | XXMix_9realisticSDXLv1.0 | 2,182 |
| 365 | YiffyMix | 4,004 |
| 366 | YiffyMixv36 | 1,038 |
| 367 | ZavyChromaXL | 4,810 |
| 368 | ZavyChromaXLv2.1 | 3,391 |
| 369 | ZavyChromaXLv3.0 | 3,655 |
| 370 | ZavyChromaXLv4.0 | 2,612 |
| 371 | ZavyChromaXLv5.0 | 3,534 |
| 372 | ZavyChromaXLv6.0 | 3,643 |
| 373 | ZavyChromaXLv7.0 | 3,843 |
| 374 | ZavyChromaXLv8.0 | 3,708 |
| 375 | ZavyChromaXLv9.0 | 4,137 |
| 377 | Zaxious_XLv2.0 | 1,158 |
| 378 | [Lah] Mysterious | Flux update | 3,822 |
| 379 | [Lah] Mysterious | Flux updatev4.0 | 1,523 |
| 380 | [PVC Style Model]Movable figure model Pony | 2,209 |
| 381 | _CHEYENNE_ | 3,716 |
| 382 | _CHEYENNE_v1.6 | 1,565 |

Table 11, continued from previous page

| Model ID | Source Name | Indexed Images |
|---|---|---|
| 384 | blue_pencil-XL | 1,752 |
| 385 | epiCPhotoGasm | 3,903 |
| 386 | epiCPhotoGasmLast Unicorn | 4,108 |
| 387 | epiCPhotoGasmUltimate Fidelity | 1,713 |
| 388 | epiCPhotoGasmZ - Universal | 2,423 |
| 389 | epiCRealism | 4,078 |
| 390 | epiCRealism XL | 3,189 |
| 392 | epiCRealismNatural Sin RC1 VAE | 3,504 |
| 394 | hash_ | 4,112 |
| 395 | hash_0685a29800 | 1,476 |
| 396 | hash_06f96f89f6 | 1,545 |
| 397 | hash_275ef623d3 | 3,236 |
| 398 | hash_3f97fdc57a | 2,085 |
| 404 | hash_bea01d51bd | 3,528 |
| 405 | hash_c161224931 | 1,665 |
| 407 | hash_d50bfd647c | 1,586 |
| 408 | hash_de5612ec57 | 1,154 |
| 412 | majicMIX realistic | 1,255 |
| 414 | majicMIX realistic v7 | 4,618 |
| 417 | qchan Mixv2.0 | 2,060 |
| 418 | qchan realistic Mixv2.0 | 1,888 |
| 419 | reweik_Uni020v20-1 | 1,097 |
| 420 | richyrichMix | 4,002 |
| 422 | richyrichMixrichyrichMix-v2.fp16 | 4,255 |
| 423 | seizaMixv2 | 1,402 |
| 424 | supashymixv3.0-lite | 1,105 |
| 425 | Realities Edge XL LIGHTNING + Turbo! | 1,323 |
| 426 | CheckpointYesMixv1.6 (original) | 1,247 |
| 427 | \| Anything XL | 3,436 |
| 428 | \| Anything XLXL | 1,110 |
| 429 | STOIQO NewReality \| FLUX, SD, XL, Lightning | 4,091 |
| 430 | STOIQO NewReality \| FLUX, SD, XL, Lightning XL 1.3 | 1,272 |
| 432 | STOIQO NewReality \| FLUX, SD, XL, Lightning XL 2.1 | 2,598 |
| 433 | STOIQO NewReality \| FLUX, SD, XL, Lightning XL 4.0 | 2,174 |
| 434 | SDXL FaeTastic | 4,511 |
| 435 | SDXL FaeTasticv1.6 | 2,389 |
| 436 | SDXL FaeTasticv2.0 | 2,518 |
| 437 | SDXL FaeTasticv24 | 4,176 |
| 438 | OpenAI 4o Image Gen 1 | 1,000 |
| 439 | Imagen 4 | 1,000 |
| 440 | Nano Banana Pro | 1,000 |
| 441 | Flux.2 Dev | 1,000 |
| 442 | Z Image Turbo | 1,000 |
| 443 | Flux.2 Klein 9B-base | 1,000 |
| 444 | Flux.2 Klein 9B | 1,000 |
| 445 | Z Image Base | 1,000 |
| 446 | Nano Banana 2 | 1,000 |
| 447 | Nano Banana | 1,000 |

