# OpenReview forum: "The Zero-Shot Illusion in the Wild: Diagnosing Boundary Failures in AIGC Detection"
_TMLR — Under review for TMLR_

### Review · Reviewer_TWrU · 2026-07-09

**Summary Of Contributions:**

This paper exposes a "zero-shot illusion" in AI-generated image detection: detectors that report strong performance on curated benchmarks (ForenSynths, GenImage) largely collapse when evaluated on a more ecologically valid distribution. To demonstrate this, the authors introduce WildGen, a large-scale benchmark constructed from 342K community-created AI images spanning 342 generative sources (primarily from Civitai) and 347K real images from 8 traceable authentic sources. On WildGen, 11 of 12 training-based detectors remain below 58% balanced accuracy, and the best training-free method reaches only 75.21% AUC. Through a diagnostic decomposition, the paper shows that the failure stems primarily from decision boundary misalignment rather than feature deficiency: frozen CLIP features support 98%+ balanced accuracy with a simple linear probe. The paper demonstrates that few-shot adaptation (as few as 18 examples per generator) can restore strong detection performance, and that detectors adapted on WildGen can transfer to existing benchmarks (GenImage, ForenSynths) via joint few-shot fine-tuning. The paper further shows that real-source identity is highly learnable (84.33% balanced accuracy), meaning the "Real" class in detection benchmarks should not be treated as homogeneous. The authors argue that rather than seeking a static universal detector, the field should pursue continuous few-shot boundary maintenance.

**Audience:**

Yes

**Audience Explanation:**

This paper will interest multiple TMLR audience segments:

- **AIGC detection researchers** (primary audience): The finding that 11 of 12 standard detectors collapse to near-chance on a more ecologically valid benchmark is a significant challenge to prevailing assumptions. The boundary-misalignment diagnosis has direct practical implications for detector design.

- **Benchmark and evaluation community** (secondary audience): The real-source bias analysis and the demonstration that "Real" is not a homogeneous class generalize beyond AIGC detection to any binary classification benchmark where one class is heterogeneous. The paper's methodology of pairing per-source analysis with aggregate metrics is a transferable evaluation design pattern.

- **AI safety and content authenticity practitioners** (tertiary audience): The paper provides concrete guidance (diversify real training data, prefer frozen-feature detectors, budget for adaptation) that is actionable even if the specific methodology requires further validation.

The paper is well-suited to TMLR's evidence-first review standard. Its contribution is empirical and diagnostic rather than methodologically novel, which aligns with TMLR's explicit policy that novelty is not required for acceptance. The benchmark fills a genuine gap, and the boundary-maintenance perspective provides a productive framing for future work.

**Broader Impact Concerns:**

The paper has significant broader impact considerations that require explicit discussion:

1. **Data provenance and consent.** The benchmark is built from scraped Civitai images. While the platform hosts publicly shared content, creators may not have anticipated or consented to inclusion in an academic benchmark. The paper should discuss opt-out mechanisms and data governance.

2. **NSFW and sensitive content.** Civitai hosts substantial adult content, and the benchmark's model catalog includes explicitly NSFW-oriented models. The paper must clarify whether such content was filtered and, if included, what safeguards accompany the benchmark release.

3. **Dual-use concerns.** Improved AIGC detection is a defensive technology, but detection benchmarks can also be used adversarially: generator developers can benchmark against WildGen to tune their models to evade detection. This dual-use nature should be discussed.

4. **Downstream bias.** If a detection system trained on WildGen inherits the benchmark's demographic, cultural, and aesthetic biases (overrepresentation of anime/fantasy aesthetics, English-only prompts, specific visual styles), it may systematically misclassify AI-generated content from underrepresented cultures, languages, or artistic traditions. The Civitai community skews toward specific demographics and content preferences; detectors deployed globally should be evaluated for fairness across these dimensions.

The authors should add a dedicated Broader Impact Statement section addressing these four concerns, with concrete mitigation strategies for each.

**Claims And Evidence:**

Yes

**Claims Explanation:**

The paper's core empirical claims are well-supported:

Detectors fail on WildGen: This is the most robust finding. With 34,200 AI + 8,000 real validation images across 342 AI sources, the evidence that most training-based detectors operate near chance level (balanced accuracy 43-58%) is statistically compelling even without explicit error bars. The per-source FPR breakdown in Table 3 provides granular evidence that failures are systematic rather than anomalous.

The failure is boundary misalignment, not feature deficiency: This claim is supported by converging evidence: t-SNE visualization (Figure 4a), a binary logistic regression probe reaching 98.15% on frozen CLIP features, a 343-way kNN classifier reaching 89.06% top-1, and the UnivFD adaptation results (97.22% at N=900, 95.77% at N=18). The paper appropriately uses multiple complementary approaches rather than relying on a single test.

- **Few-shot adaptation is effective**: The dose-response relationship across N ∈ {900, 90, 45, 18} in Table 6, with monotonic performance degradation, provides credible evidence. The strong performance at N=18 is surprising and deserves the additional scrutiny requested in Weaknesses #5-6, but the qualitative conclusion that few-shot adaptation works is well-supported.

Real-source bias matters: The 8-way real-source classifier (84.33% balanced accuracy, Table 10) and per-source FPR analysis (Table 3) provide clear, direct evidence.

The claims that need qualification are concentrated in the abstract (Weakness #2) and the concluding prescriptive statement (Weakness #3). The body text generally contains appropriate caveats that the abstract strips away. The paper would be substantially strengthened by: (1) adding confidence intervals to all main tables, (2) clarifying the deterministic split mechanism, (3) reporting train-set performance for adaptation experiments, and (4) adding an ethics statement.

**Requested Changes:**

### Critical to Acceptance

1. **Add an ethics statement (Weakness #1).** This must discuss: (a) the provenance of Civitai-sourced images and platform ToS compliance, (b) whether and how NSFW content was handled, (c) creator consent and opt-out mechanisms, (d) privacy implications of LoRA-based models fine-tuned on individuals' faces, (e) licensing/attribution for real-image sources, and (f) downstream bias risks. A data governance plan for the released benchmark should be outlined.

2. **Revise the abstract to accurately reflect the evidence (Weakness #2).** Replace "unseen models" with "held-out sources within WildGen" or equivalent. Clarify that cross-benchmark adaptation uses joint (WildGen + target) training, not target-only few-shot data. The concluding prescriptive statement ("a more effective and realistic path") should be softened to reflect that the evidence demonstrates feasibility, not comparative superiority over alternatives.

3. **Revise the "long-tail" framing or the benchmark construction methodology (Weakness #3).** Either (a) drop the "long-tail" language and accurately describe WildGen as capturing the popular head of the Civitai distribution, or (b) include the 105 excluded sub-1,000-image sources in a separate evaluation to demonstrate generalization to truly rare models. Option (a) is preferred as it requires only text changes.

### Would Strengthen the Work

4. **Add confidence intervals to all main tables (Weakness #4).** Clopper-Pearson binomial intervals for balanced accuracy, bootstrap CIs for macro-averaged AUC. This requires no additional computation beyond what has already been done.

5. **Clarify the deterministic split mechanism (Weakness #5).** Specify the selection rule explicitly (random seed? chronological? hash-based?). If chronological, either re-split randomly or demonstrate no temporal confounding.

6. **Report train-set performance for adaptation experiments (Weakness #6).** Adding training accuracy alongside validation accuracy in Table 6 would directly address the overfitting concern for CNNDetection at N=18.

7. **Add a "Scope and Limitations" subsection to Section 3 (Weakness #7).** Characterize the Civitai distribution: what types of AI images are overrepresented (anime, furry, photorealistic, fantasy), underrepresented (corporate/enterprise AI tools, non-English prompt cultures), and absent (DALL-E, Midjourney, Adobe Firefly). Quantify the architectural concentration (SD1.5 vs. SDXL vs. FLUX vs. other base families).

8. **Discuss the deployment circularity of few-shot adaptation (Weakness #9).** Add a paragraph to Section 5 or the Conclusion describing how source identification for adaptation could work in practice, and what the cold-start limitations are.

9. **Quantify architectural concentration among the 342 sources.** A one-paragraph summary (e.g., "approximately X% are SD1.5-based, Y% are SDXL/Pony-based, Z% are FLUX-based, W% are other") would help readers calibrate the diversity claims.

10. **Add a note on statistical power.** A sentence or footnote acknowledging that the large validation set provides implicit variance reduction (effective SE approximately 0.3 percentage points for aggregate balanced accuracy) would preempt methodological concerns about missing error bars for qualitative claims.

11. **Consider whether Appendix B (LMM case study) adds or detracts.** The authors' own disclaimer that it is "not a systematic benchmark" and "should not be treated as quantitative evidence" raises the question of whether it belongs in the paper at all. Moving it to supplementary-only or expanding it to a systematic study would be preferable to the current ambiguous status.

---

### Review · Reviewer_CCDD · 2026-07-11

**Summary Of Contributions:**

The paper evaluates a broad set of training-based and training-free AIGC detectors and finds that many methods reporting strong performance on established benchmarks fail substantially on WildGen. In particular, several trained detectors collapse toward predicting one class, while training-free approaches show strong source-dependent variation. The paper refers to this discrepancy as the “zero-shot illusion.”

The authors further argue that these failures are primarily caused by decision-boundary misalignment rather than insufficient representation quality. This argument is supported by experiments showing that frozen CLIP features can achieve high performance after fitting a new linear classifier on WildGen. The paper additionally studies few-shot adaptation, generator/source identification, real-source bias, cross-benchmark transfer to GenImage and ForenSynths, and robustness to resizing and JPEG compression.

**Additional Comments:**

The paper is generally readable and the central motivation is clearly presented.
Figure 4’s t-SNE visualization should not be treated as quantitative evidence. It would be more informative to report linear separability and nearest-neighbor results across multiple seeds, along with controlled baselines.
Figure 6 appears to show one individual source with AUC below random. The caption should make explicit that this is a selected case rather than the overall RIGID result.
The comparison between training-based balanced accuracy and training-free AUC/AP is appropriately noted as non-comparable, but a unified evaluation would still be helpful. For example, all methods could be evaluated using AUC, AP, and calibrated balanced accuracy under a fixed calibration protocol.

**Audience:**

Yes

**Audience Explanation:**

The paper addresses a problem that should be of substantial interest to researchers working on AI-generated media detection, distribution shift, dataset design, multimodal representation learning, digital forensics, and trustworthy machine learning.

The finding that detector performance can collapse when moving from curated benchmarks to a heterogeneous community-generated distribution is important, even if the precise mechanism remains unresolved. In particular, the paper’s treatment of the real-image class is valuable. Many AIGC detection studies diversify the synthetic generators while keeping the real distribution fixed, which can encourage models to learn dataset identity rather than authenticity. The source-specific false-positive results and the strong predictability of real-source identity provide a useful warning for future benchmark construction.

**Broader Impact Concerns:**

The paper addresses a socially important problem, but the benchmark raises several ethical and governance issues that should be discussed more explicitly.

First, a substantial portion of the dataset appears to be collected from a community platform and includes model categories or names suggesting adult or sensitive content. The authors should state whether NSFW material, sexualized depictions, depictions of minors, copyrighted characters, personal likenesses, or other sensitive categories are included, and what filtering and access controls will be applied.

Second, the paper should clarify the legal and licensing basis for redistributing the images. Public accessibility on a platform does not necessarily imply permission for bulk redistribution. It may be safer to release URLs, hashes, metadata, or a controlled download tool rather than repackaging all image files, depending on the relevant licenses and platform terms.

Third, benchmark images may contain personal data, watermarks, usernames, artist signatures, or identifiable individuals. The paper should describe privacy review, metadata sanitization, opt-out mechanisms, and takedown procedures.

**Claims And Evidence:**

Yes

**Claims Explanation:**

The paper provides convincing evidence for the narrower empirical claim that many existing AIGC detectors perform poorly when evaluated on the WildGen distribution. Table 3 shows severe and source-dependent failures across a broad range of trained detectors, while Table 4 demonstrates that the evaluated training-free approaches also remain far from reliable. The real-source analysis is similarly informative: an eight-way classifier predicts the originating real dataset with 84.33% balanced accuracy, and several detectors exhibit highly concentrated false-positive rates on particular real datasets. These results convincingly demonstrate that benchmark composition, including the real-image distribution, strongly affects measured detection performance.

**Requested Changes:**

1. Add generator-disjoint and preferably family-disjoint evaluations.

The most important missing experiment is a split in which complete generators are held out from training. The current adaptation protocol uses N samples from every evaluated generator and therefore measures same-generator adaptation.

2. Reframe or more rigorously validate the “boundary misalignment” claim.

The current linear-probe result establishes target-distribution separability, but not that the features encode transferable forensic evidence or that boundary misalignment is the primary cause of detector failure.

3. Perform rigorous deduplication and leakage analysis.
A source-family graph or manually curated checkpoint lineage mapping would substantially strengthen the benchmark.


4 Report repeated-run uncertainty.

The few-shot results use as few as 18 samples per generator but are reported without variance. Results should be averaged across multiple independently sampled training subsets and random seeds. Confidence intervals are necessary to determine whether differences between methods and data regimes are meaningful.

5 Clarify benchmark chronology.

A temporal split would better approximate deployment. Training on older generators and testing on generators or images released later would provide a much stronger evaluation of evolving detector boundaries than a random split within each generator.

---

### Review · Reviewer_cUo2 · 2026-07-18

**Summary Of Contributions:**

Detecting AI images is a difficult and important problem for which many methods have been developed. Existing AI-generated content (AIGC) detection benchmarks do not seem to capture the complexity of AIGC detection “in-real life.” This paper proposes a new benchmark, WildGen, that addresses this concern by (1) sourcing AIGC from an online platform, where images come from a wider variety of architectures, and (2) sourcing real-world images from eight different sources (most of the existing benchmarks source from 1 to 3 sources).
They find that the training-free method RIGID (which uses the stability of the embedding as a proxy) has the highest zero-shot accuracy on their benchmark, while training-based methods have poor zero-shot but strong few-shot adaptation performance (for AI-vs-real classification, goes from 58% zero-shot to 95% with 18 examples per generator).

**Additional Comments:**

In the context of real-vs-AIGC detection: should drawn / animated / CGI images typically be included in the “real image” class?

Did you consider AI-generated-images prompted adversarially, like telling the AI what the common signs of AI-generated images are and to “try to make it impossible to distinguish with a real image”?

There is a repetitive sentence structure in this paper which I find curious, at least in the last few pages.
- "We therefore treat real-source diversity as a core benchmark rather than a cosmetic data expansion"
- "We therefore report real-source bias and ratio sensitivity as controls, and avoid interpreting the Real class as a monolithic concept"
- "We therefore treat environmental robustness as supporting evidence: distribution shift and real-source confounding remain the more important failure modes"
- "We therefore argue for reporting the distributional boundary within which a detector is valid, and hope WildGen offers a reproducible foundation as generators and real-image sources evolve"
The authors might consider varying their sentence structure.

**Audience:**

Yes

**Audience Explanation:**

I think some members of TMLR’s audience would be interested in knowing the existence of this new benchmark and what it shows about the narrowness of previous benchmarks. I am not sure how interested the relevant community would be about the findings of the analysis, re: the efficacy of few-shot over zero-shot adaptation in this context.

**Broader Impact Concerns:**

I am somewhat concerned about the potential presence of NSFW images in the benchmark / images which are not reflective of the values of our community (7 of the base models have the word “hentai” in their names, and another 7 have the word “NSFW” in them). I strongly suggest that this is addressed explicitly in the paper, perhaps in a broader impact statement.

**Claims And Evidence:**

Yes

**Claims Explanation:**

The main claim of the paper is that their new, more comprehensive/realistic benchmark for AIGC is hard for existing models to tackle zero-shot, but quite amenable to few-shot adaptation. The experiments and evidence are clear and convincing, though I do wonder what the few-shot performance looks like for 1 < N < 18 (the smallest-N few-shot performance recorded was for N=18). A plot of N-shot performance vs N would be interesting to compare across the different models.

One claim/implication in this paper of which I am somewhat skeptical is this concept of the “zero-shot illusion”: that there is this mistaken impression in the literature that AIGC can be tackled zero shot and their paper claims or at least implies to be the first to disprove it. It strikes me as a straw man argument.

**Requested Changes:**

See the concerns/requested changes brought up in other questions, re:

- the framing of the "zero-shot illusion" as a straw man
- the presence of NSFW images
- more granular study of the few-shot adaptation performance

The majority of my requested changes involve the figures.

- I think it would be more effective if Figure 1 showcased more than 1 example. I think 6-8 diverse examples would be reasonable, perhaps AI-real pairs paired by topic.
- I am not sure what the message of Figure 2 is. Some of the words make me bring the concern of NSFW images to mind.
- I am also confused about the message of Figure 3. If both AI generated images and real images exhibit low resolution, how does this enable shortcuts for learning?
- Tables 4,5 and Figure 6: I think it would be more meaningful to show the performance numbers of training-free and training-based methods next to each other rather than in separate tables/figures. Also, consider using bar graphs instead of these numerical tables.
- Figure 5: it is not clear what the visual difference between the Pixel Art XL and qchan Mix residuals tell us about the difficulty of distinguishing these from the real data. It might be helpful to show residual between this real image and another real image, and show that it looks more like the Pixel Art XL residual (though why focus on this in the first place if it's not the contribution of this paper).
- The alleged purpose of the t-SNE projection in Figure 4a is to show that there is real-vs-AI separation. First of all, the legend labels are illegible. Secondly, there are stronger, quantitative ways of showing this separation. As for visualization, did you consider spectral clustering?